# Developing an organizational capacity assessment tool and capacity-building package for the National Center for Prevention and Control of Noncommunicable Diseases in Iran

**Ahad Bakhtiari**[1,2], **Amirhossein Takian**[1,2,3]*, **Afshin Ostovar**[4], **Masoud Behzadifar**[5], **Efat Mohamadi**[1], **Maryam Ramezani**[1,2]

1 Health Equity Research Centre (HERC), Tehran University of Medical Sciences, Tehran, Iran,
2 Department of Health Management, Policy and Economics, School of Public Health, Tehran University of Medical Sciences, Tehran, Iran, 3 Department of Global Health and Public Policy, School of Public Health, Tehran University of Medical Sciences, Tehran, Iran, 4 Osteoporosis Research Center, Endocrinology and Metabolism Clinical Sciences Institute, Tehran University of Medical Sciences, Tehran, Iran, 5 Social Determinants of Health Research Center, Lorestan University of Medical Sciences, Khorramabad, Iran

* Takian@tums.ac.ir

## Abstract

Non-communicable diseases represent 71% of all deaths worldwide. In 2015, Sustainable Development Goals, including target 3.4 of SDGs, were seated on the world agenda; "By 2030, reduce premature mortality from NCDs by one-third. More than half of the world's countries are not on track to reach SDG 3.4, and the COVID-19 crisis has hampered the delivery of essential NCD services globally, which means the premature death of millions of people and indicates the need for capacity building for health systems. We designed a tool to measure the capacity of the National Center for Non-Communicable Disease and then presented the proposed policy package to enhance the national center's organizational capacity. The data for this explanatory sequential mixed method study was collected using quantitative and qualitative approaches between February 2020 and December 2021. The tool for assessing organizational capacity for NCDs was developed, and its validity and reliability were measured. The developed tool assessed the organizational capacity by evaluating NCNCD's managers and experts. Following the quantitative phase, a qualitative phase focused on the low-capacity points revealed by the tool. The causes of low capacity were investigated, as well as potential interventions to improve capacity. The developed tool comprises six main domains and eighteen subdomains, including (Governance, Organizational Management, Human Resources Management, Financial Management, Program Management, and Relations Management) which verified validity and reliability. In seven separate National Center for Non-Communicable Disease units, the organizational capacity was measured using the designed tool. (Cardiovascular disease and hypertension; diabetes; chronic respiratory disease; obesity and physical activity; tobacco and alcohol; nutrition; and cancers). The organizational management dimensions and the sub-dimensions of the organizational structure of the Ministry of Health and Medical Education and units affiliated with

**Data Availability Statement:** All relevant data are within the paper and its Supporting information files.

**Funding:** The authors received no specific funding for this work.

**Competing interests:** AT and AO are members of the INCDC at the MoHME- Iran. We (AT and AO) have no conflict of interest to disclose. AB, EM, MB, and MR declare that they have no competing interests.

**Abbreviations:** DALYs, Disability-adjusted life years; HTTP, Health transformation plan; I-CVI, Item-level Content Validity Index; INCDC, Iran NCDs Committee; MoHME, Ministry of Health and Medical Education; NCDs, Non-communicable diseases; NCNCD, National Center for Non-Communicable Disease; OCAT, Organizational Capacity Assessment Tool; PHSSR, Public Health Services and Systems Research; SCHFS, Supreme Council of Health and Food Security; S-CVI, Content Validity Index for Scales; SDGs, Sustainable Development Goals; SIB, Integrated health system.

the national center, in all cases, were almost one of the main challenges that affected the country's capacity to fight against NCDs. However, all units had a relatively good situation in terms of governance (mission statement, vision, and written strategic plan). The content analysis of experts' opinions on the low-capacity subdomains highlighted challenges and recommended capacity-building interventions. Transparency in methods and processes is necessary to allocate funding among various health programs and evaluate their effects through cost-effectiveness indicators. This study identified weak points or areas where capacity building is required. The root causes of low capacity and interventions to build capacity are listed in each dimension of the tool. Some of the proposed interventions, such as strengthening organizational structures, have the potential to impact other domains. Improving organizational capacity for NCDs can assist countries to achieve national and global goals with greater efficiency.

## Introduction

Non-communicable diseases (NCDs) represent 71% of all deaths worldwide and kill 41 million bodies each year; the four top killers that together account for more than 80% of all premature NCD deaths include cardiovascular diseases (17·9 million deaths annually), cancers (9 million), respiratory diseases (3·9 million), and diabetes (1·6 million) [1]. NCDs killed 287000 people in Iran in 2016 [2]. Rising trends in the number of associated death and disability-adjusted life years (DALYs) during the past decades have played an alarm for policymakers in the world and Iran [3]. In 2014, the Ministry of Health and Medical Education (MoHME) of Iran and the National Center for Non-Communicable Disease (NCNCD) began significant reforms to the health system; IraPEN is part of the national health transformation plan (HTP) to provide universal health coverage, including access to NCD prevention and care, and mental health services [4].

The 25 × 25 strategy for the burden of NCDs was reinstated in 2015 with the new Global Agenda, "Sustainable development goals" (SDGs) especially target 3.4 of SDGs; "By 2030, reduce by one-third premature mortality from NCDs through prevention and treatment and promote mental health and well-being" [5]. Several guides and interventions have been considered and recommended to achieve target 3.4, including Package of Essential Noncommunicable (WHO-PEN) Disease Interventions for Primary Health Care in Low-Resource Settings [6], Ira-PEN [4], Best Buys [7], the relation between NCDs and universal health coverage (UHC) [8]. In a complementary study, we evaluated the national NCDs program and prioritized best buys and other WHO-recommended interventions in Iran; the results show that the WHO recommendations for NCDs management are included in the national program [9]. We also identified and analyzed the key stakeholders in implementing the WHO-recommended interventions in Iran, which will assist policymakers in developing inter-sectoral cooperation [10].

More than half of the world's countries are not on track to reach SDG 3.4 [11, 12], and the COVID-19 crisis interrupted delivering healthcare services for hypertension, diabetes, cancer, and cardiovascular disease in Iran; also, caused service delivery for blood pressure, diabetes, cancer, and cardiovascular disease was disrupted by 22 to 66 percent in 163 countries; [13, 14], which means the premature death of millions of people and highlighting the need for capacity building [15].

For the better direction of the public health sector, Public Health Services and Systems Research (PHSSR) was introduced in 2001 [16] but focused more on performance and

standards, and key processes, the conceptual framework of PHSSR refers to "capacity," but there is not much to be said about its definition and how to measure capacity [17]. Along with the development of the general framework for capacity assessment, researchers' and studies' focus on organizational capacity and related components has grown over time. The specific framework for assessing organizational capacity for diseases such as AIDS was developed by public health researchers [18].

Definitions of Capacity Building for disease programs can be defined as follows:

"Capacity building is any action that improves the effectiveness of individuals, organizations, networks, or systems—including organizational and financial stability, program service delivery, program quality, and growth" or "Capacity building is a long-term process that improves the ability of an individual, group, organization, or ecosystem to create positive change and perform better to improve public health results" [19].

Capacity: the ability or power of an organization to apply its skills, assets, and resources to achieve its goals. An organization with high organizational capacity will have higher speed, power, and quality in attending to its goals. Governance, the hierarchy of authority, organizational structure, human resource management, financing, resource allocation, administrative communications, legislation, and inter-sectoral coordination are all affected by organizational capacity. Improving their management equals better management for the health of the population.

Organizational capacity assessment for NCDs is one of the WHO-recommended interventions that, according to our previous research, was not on the national agenda [9]. On the other hand, other capacity measurement techniques weren't appropriate for Iran's context. For instance, Iran has provided a positive response to the question of whether there is a national institution for NCDs in the WHO's capacity measurement tool. But it doesn't mention the fact that conditions like respiratory diseases and physical inactivity lack formal organizational structures and that multiple parallel offices are concurrently engaged in their own curative and public health deputies. According to this situation, this study aimed to assess the organizational capacity assessment of NCNCD to strengthen the NCNCD organizational capacity and improve the administration of NCDs in Iran. We designed a tool to measure the capacity of NCNCD and then presented the proposed policy package to enhance the NCNCD organizational capacity.

## Methods

The data for this explanatory sequential mixed method study was collected using quantitative and qualitative approaches between February 2020 and December 2021. In the initial phases, the organizational capacity assessment tool was developed, its validity and reliability were measured, the organizational capacity of NCNCD was assessed. Following the quantitative phase, a qualitative phase focused on the low capacity points revealed by the tool. The causes of low capacity were investigated, as well as potential interventions to improve capacity.

### 1. Designing organizational capacity assessment tool

**1.1 Literature review and understanding of the organization that will be evaluated.**  To become familiar with organizational capacity building, the literature reviewed in various fields, including public health. Recommendations, dimensions, methods, and steps for capacity assessments/building in literature were reviewed and summarized. Next, the study team became more familiar with the four major NCDs and related risk factors. The national documents and policies, as well as global documents, policies, and recommendations reviewed (Box A in S1 Appendix) [9].

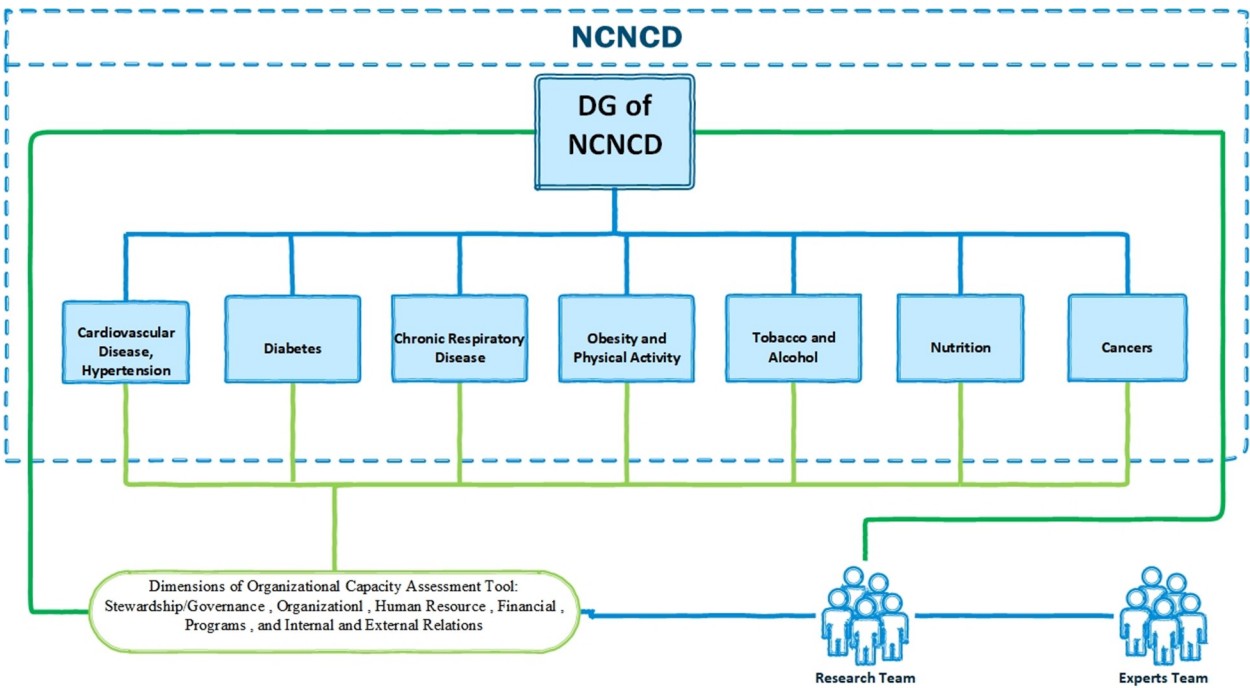

**Fig 1. Selected subjects for capacity assessment.**

**1.2 Understanding the NCNCD as a targeted organization for capacity building.** The initial collaboration with the NCNCD's director was conducted, and the structure and position of the NCNCD within the (MoHME) were reviewed. An NCNCD assistant was also chosen as the study team's communication link. The Ministry of Health's overall structure was also reviewed to discover other relevant departments.

As the leading NCDs and related risk factors, seven subjects for capacity assessment and capacity building were selected according to Appendix 3 of the WHO Global NCD Action Plan [18]; based on capacity assessment results on these seven subjects (Fig 1), the primary organizational barriers and challenges surrounding the NCNCD were identified.

## 2. Designing tool (Organizational Capacity Assessment Tool, OCAT)

The tool's basic foundation was designed based on a comprehensive review of similar tools and the relevant literature (PubMed, Scopus, and Web of Science) [6, 19–31] (Box B in S1 Appendix). we have divided assessment questions into six general domains that make up the main parts of the tool (Fig 2). Based on the relevant literature and experts' opinions, the subdomains were designed to measure the extent of NCNCD's control and authority over those domains; while the WHO questionnaire measures the presence or absence of factors in most cases, the tool measures NCNCD's authority over the designed sub/domains. Each domain's questions were designed based on reviewing related studies and opinions of the study team and the expert team number one (Table 1 in S1 Appendix). The relevant expert team discussed and decided on the tool's questions and structure. Three situations were labeled as unacceptable, requiring revision, and acceptable for the sub-dimension. The questions were revised continuously until they were all in an acceptable situation. To minimize measurement error, the researchers used short, simple sentences and non-specialized words as answers.

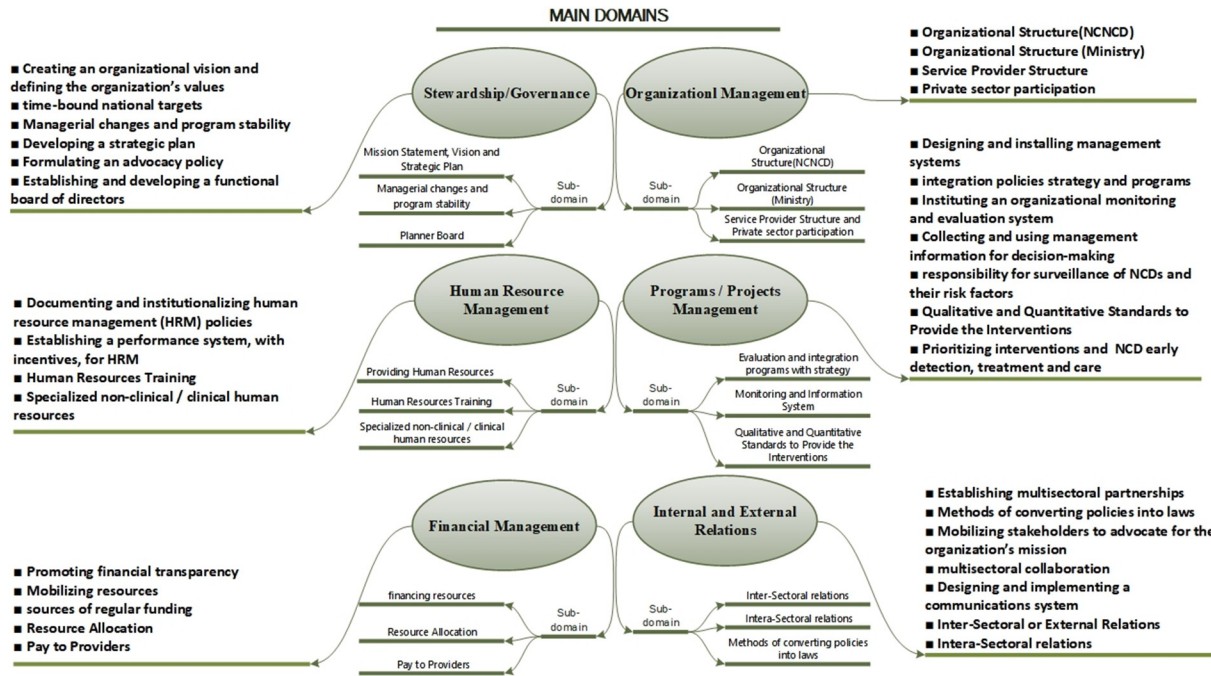

**Fig 2. Main domains and sub-domains considered for capacity assessment.**

**2.1 Tool's validity and reliability.** We considered expert team number two for the tool's validity and reliability (18 experts), with 12 of them selected purposely with academic expertise and experience in the field of organizational aspects to examine the tool's validity and six experts to evaluate the tool's reliability based on their familiarity with the NCNCD to determine the inter-rater agreement between their scores and the managers of the NCNCD (Table 2 in S1 Appendix). Testing content validity can be done in a variety of approaches. The methodology used in this study included both cognitive approaches and empirical methods to determine the content validity ratio (CVR) and content validity index (CVI). Following these steps, cognitive interviews with six NCNCD experts were established. We used the Kappa statistic to evaluate the reliability. Statistical analysis for reliability testing was conducted using R version 3.6.2 (34), package DescTools (35). For more details about tools, validity, and reliability, please see S1 Appendix.

## 3. Data collection: The process of capacity assessment and capacity building

We considered three general steps of the capacity-building process (Fig 3), including 1. Assessment (and Reassessment), 2. Identify Solutions and Action Planning, and 3. Modification Domains and implementation.

**3.1 Capacity assessment.** The experts from each office with a connection to the areas under investigation received the finalized tool. They received the verbal and written guidance they required and had access to the NCNCD facilitator if they had any questions. Their colleagues rechecked the scored tool after it had been scored. To check and finalize the scoring of the tool, a meeting was conducted with the experts and managers of all 7 areas and the head of NCNCD.

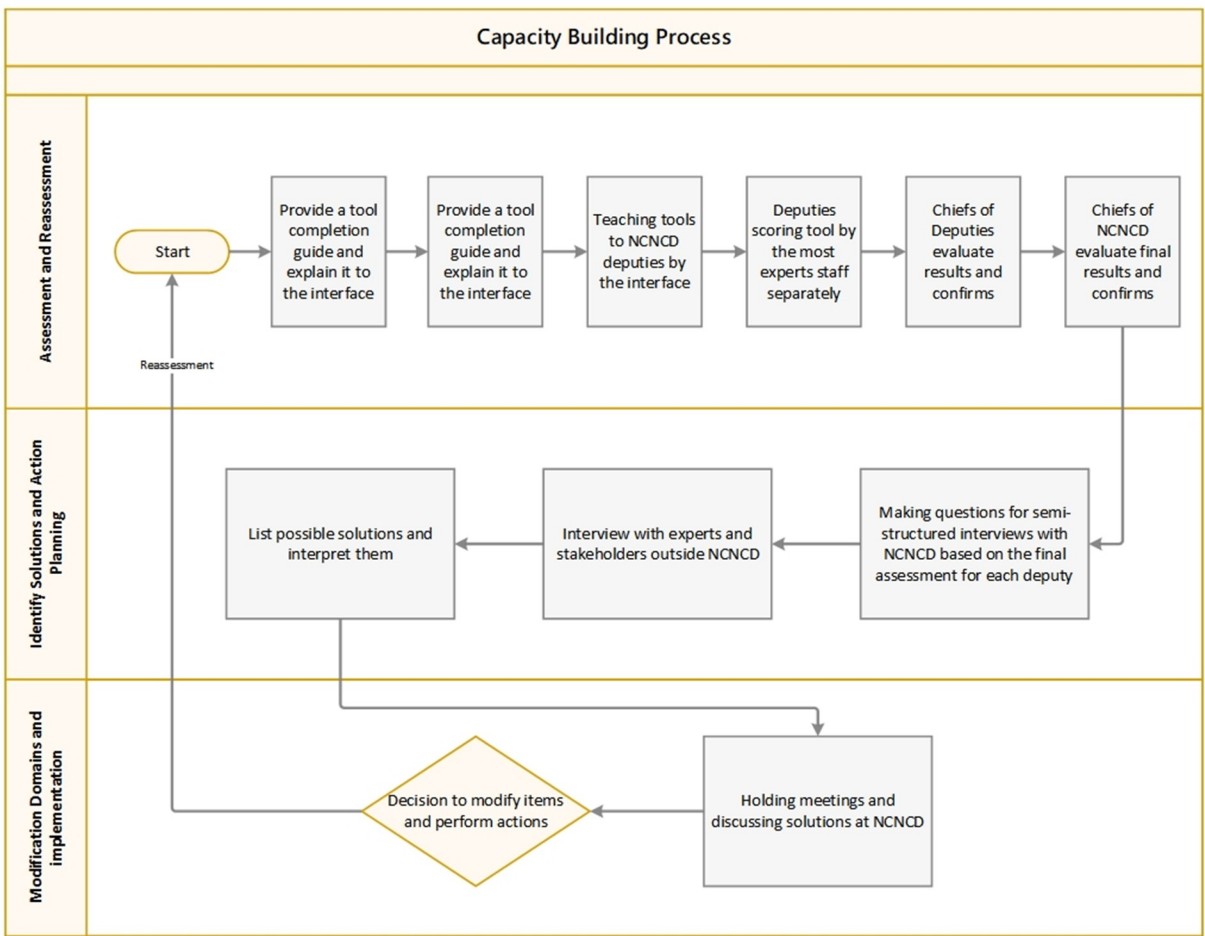

**Fig 3. The general steps of the capacity-building process.**

The sum of three questions will determine the score for each domain; After scoring the OCAT, weaknesses were identified in each of the seven subjects or units; also, common weaknesses were identified. After passing a familiarity course conducted by the interface (NCNCD assistant), seven experts from seven units of NCNCD rated the designed tool. To ensure the accuracy of scoring, the rated tools were evaluated by seven unit managers and the chief of NCNCD.

A score of $X \leq 2$ on a scale of 4 in the sub-domain and a score of $X \leq 8$ on a scale of 12 in total scores was considered in the tool analysis to be a point that needed capacity building.

## 4. Qualitative phase: Challenges and recommended capacity-building interventions

**4.1 Interviews with the NCNCD and the third expert team.** The common weaknesses among the seven deputies are the most important points we explored through interviewing experts. Semi-structured face-to-face interviews with purposefully selected experts were conducted (AB—Ph.D. and MB–Ph.D., both have experience in national studies with a qualitative approach) to identify the roots of the problems and develop capacity-building recommended interventions. We chose experts purposively and used snowball techniques. (MoHME's

departments are relevant to seven selected subjects or units belonging to NCNCD, asked NCNCD-unite managers (tool raters) to introduce influential stakeholders in relevant subjects) (Table 3 in S1 Appendix).

The study's aims were described to the interviewees before the interview, and they also received information on the topic in question. The interviews, which lasted between 30 and 80 minutes each, were conducted between August 2020 and March 20201 to ensure data saturation. The interviews were all conducted at the interviewees' workplaces. The interviews were conducted using an interview guide that was created based on the domains of the tool and recognized weaknesses. Interviews were recorded, and transcribed verbatim; the data were analyzed using Framework content analysis. The steps included in framework analysis are Familiarization or reviewing the interviews as well as the knowledge gained in the previous phases; Identifying a thematic framework or the domains and subdomains of the tool; Indexing or Line by line coding of interviews; Charting. arranging codes to affiliated subdomains; and Mapping and interpretation [32]. Framework content analysis started with independent work from AB and MB and disagreements and ambiguities were brought up and addressed.

To ensure the quality of the qualitative phase, numerous steps were taken, including using an interview guide, selecting a varied range of interviewees, replicating data and statements during the interviews, doing multiple analyses, and validating the instrument's dimensions. The study received ethical approval from the Ethical Committee of the Tehran University of Medical Sciences (Approval number 9321460002).

## 5. Modification domains and implementation

Based on the content analysis opinions of NCNCD management and experts and the views of experts from outside the NCNCD, a package of interventions in six domains was identified. The proposed interventions were presented to the NCNCD as a recommended capacity-building policy package.

## Results

A capacity assessment tool is a tool that can be used to assess an organization's capacity. The designed tool was used in seven different sections of NCNCD. (1. Cardiovascular disease and hypertension; 2. Diabetes; 3. Chronic Respiratory Disease; 4. Obesity and Physical Activity; 5. Tobacco and Alcohol; 6. Nutrition (salt, sugar, fat, fruit, and vegetable); and 7. Cancers). According to Lawshe's content validity ratio [33], the CVR was calculated in all cases and was above the minimum value of 0.56, and the S-CVI was equal to 0.903. (Tables 6–11 in S1 Appendix). The kappa values for the three arias we tested show moderate (0.4–0.6) to strong (0.6 and higher) interrater agreement (Table 5 in S1 Appendix). This indicates that the tool and its application methodology meet the reliability standard. We reasoned that if both raters were trustworthy, they should report the same organizational capacity level. Kappa statistics have confirmed this hypothesis.

The radar chart provides a quick snapshot of a unit's overall organizational capacity status. Fig 4 depicts the results of capacity assessments for different subjects; for each topic, six domains are scored; this evaluation is also performed for subdomains (Figs 3 to 9 in S1 Appendix). Many of the assessed areas, as depicted in Fig 4, have weaknesses in the organizational management, relations, and financial management dimensions. Relations, followed by financial management, are the two main domains for developing capacity in the field of cardiovascular disease and hypertension. It should be noted that this flaw may have its roots in other areas, such as the organizational structure. Organizational management is the main domain

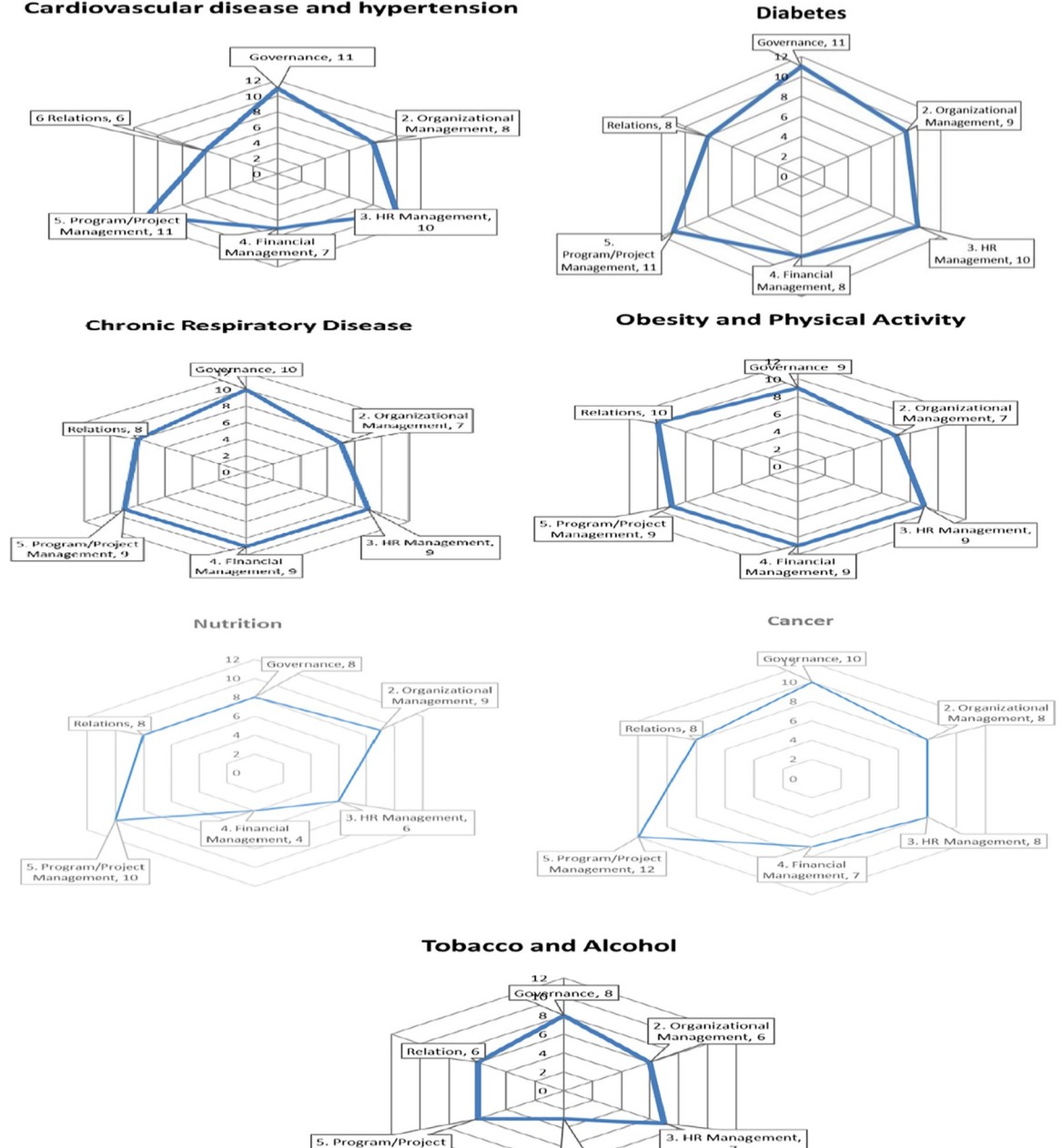

**Fig 4. The capacity assessment results.** 1. Cardiovascular disease and hypertension; 2. Diabetes; 3. Chronic Respiratory Disease; 4. Obesity and Physical Activity; 5. Tobacco and Alcohol; 6. Nutrition; 7. Cancers.

that requires capacity building in areas 3. Chronic Respiratory Disease, 4. Obesity and Physical Activity, and 5. Tobacco and Alcohol. Financial management has the greatest need for organizational capacity building in the area of nutrition and cancer.

After the tools identify the subdomains that require capacity building, the qualitative study is used to identify the causes of weakness and the best interventions for capacity building in

each sub-dimension. The six main themes (domains) and 18 sub-themes (subdomains) were utilized for arranging 438 codes. The results of the analysis of the interviews in each of the main dimensions are as follows.

## 1. Stewardship/governance

Cardiovascular diseases and hypertension, diabetes, chronic respiratory diseases, and cancers have higher scores (11, 11, 10, and 10, respectively) in the Governance domain because they are directly within the NCNCD structure (NCNCD is within the deputy for public health). In contrast, the parallel governance system exists in the deputy for curative affairs, which has created challenges for integrated governance. The tobacco control secretariat and the community nutrition improvement office play a crucial role in the governance of nutrition, tobacco, and alcohol. Both positions belong to the deputy for public health, which has made cooperation with the NCNCD for governance more coherent than the deputy for curative affairs.

As one of the most comprehensive and successful programs, Iran's National NCDs Program has clearly established the country's path. Under normal conditions, its comprehensiveness is a strength, but it might become a weakness with the economic sanctions and the COVID-19 crisis. *"Another concern with our planning is idealism. We suddenly see that a substantial package has been developed that will be difficult to implement, particularly in light of the sanctions that we met throughout IraPEN's implementation."* (P.4).

Since the National Committee for NCDs was established in 2015 under the MoHME, governance of NCDs has been reinforced. The committee's diversity of specialties has separated program direction from personal perspectives.

> *"There are specific concepts involved in preventing NCDs, and structural changes do not occur" (P.2) "and we will keep to our original plans (P.20), even though "switch in governments has caused significant challenges"*
>
> *(P.23).*

Many departments are policy-making for NCDs and related risk factors requiring NCNCD policy approval. NCNCD leaders are emphasizing increased collaborative partnerships.

> *"It is irrational to have an office within the NCNCD for any risk factor. However, the working groups should include a diverse range of stakeholders, and the NCNCD should only proceed with the coordination, finalization, and approval of policies and programs"*
>
> *(P.4)*

The governance is dual, parallel, and distinct. In the public health and curative affairs of MoHME, governance is often separate, and policies are inconsistent. The NCDs governance has been substantially challenged by the unclear position of two decades of poor implementation of the family physician program and the referral system because these are the platforms for conducting the programs.

> *"A national director of a disease must not only be accountable for public health, but also have the authority to supervise treatment" (P.10). "If you review the organization structure, the human resources, and the overview, you should give NCDs as much consideration as necessary." Additionally, the structure should be modified in this manner.*
>
> *(P.13)*

Provincial needs and priorities in NCDs governance aria would vary according to Iran's vast geographical, cultural, and social context. Provincial policymakers must be able to modify national programs based on these variances, but not all provinces have this capability. *"Not all provinces have the capacity to modify the programs" (P.2)*.

The executive body will show opposition during the implementation phase if it does not actively participate in the planning phase. Experts also identified written or verbal commitments as a concern. *"Although this document was signed by high-ranking officials who committed their support, we did not receive it in practice" (P.5)*.

Furthermore, the outcome of long-term preventative activities is determined, and managers are proclive to pursue steps that deliver immediate results. *"The advantages of preventive work take time to show up, but managers desire to see the rewards of their efforts right away." (P.8)*.

For decades, national macro-plans, like the family physician, that affect other programs like an umbrella, have remained undefined, putting NCDs planning in a state of flux. *"We need to decide whether or not we want to use the family physician and specify the task. The program's unpredictability posed numerous challenges to developing and executing diabetes-related activities." (P.9)*

Multi-sectoral Governance for health has been forgotten. *"Vital nutritional topics of the country are left, i.e., the Ministry of Health should be strongly involved in what is subsidized, for example, sugar subsidy as a risk factor for NCDs or subsidizing saturated oils" (p.13) "The Ministry of Health has some influence on food safety, which is food health, but its role in food security is much lower than it should be" (P.14)*

Recommendation

- Capacity Building for NCDs provincial teams at medical universities to guide and modify national programs based on the context of provincial relevant indicators.

- Strengthening the Ministry of Health's intra-inter-sectoral relations on NCNCD and their risk factors for coherence in national planning and policies, as well as forming joint working groups among the Ministry of Health's departments and deputies, and increasing the executive body's participation in updating national programs.

- Using disease burden and cost reports to instill the necessity of an NCDs program and its promotion within the scientific community and national and provincial decision-makers.

- Clarifying the future path of the Iranian health system's governance for managers and policymakers; determining national inclusion programs like family physicians and referral systems that greatly influence planners' decisions.

- Examine how disease governors can be integrated into the public health and curative departments and explore similar scenarios.

- Strengthen the Ministry of Health's governance in health-related concerns, including the Ministry of Commerce, the Ministry of Welfare, and Nutrition.

## 2. Organizational management

Comparatively, the organizational structure of the examined units in cardiovascular diseases, diabetes, cancers, and nutrition is more formally defined and coherent than the organizational structure in chronic respiratory diseases, tobacco and alcohol, and physical activity. The Ministry of Health's organizational structure has led to separating treatment and public health into two distinct departments, and opinions regarding the optimal structure and possible solutions vary. The Deputy Public Health has established one of the most extensive PHC networks in

the world, especially in rural areas. Following the 2014 health transformation plan, this structure was strengthened, particularly in urban areas where active PHC service providers had no comprehensive coverage.

After implementing the HTP, providing active service packages, especially NCDs, became more attainable. *"Consider a patient with three chronic diseases. If he/she must visit a separate center for each and cannot afford to do so, rural and urban comprehensive health centers can be fixed this concern" (P.9).* The MoHME's network management center, which makes decisions about the Primary health packages that available in PHC facilities, is a key component of the structure and is essential to the efficiency of the current structure.

The Supreme Council of Health and Food Security (SCHFS) is the principal office of MoHME in promoting inter-sectoral cooperation. The establishment of this council's secretariat within the structure of the Ministry of Health has facilitated the development of inter-sectoral interventions.

> *"In the case of expert nutrition, which has been used in the PHC centers, it is a good initiative, but it cannot do much; the food policy in the country must be addressed, which is not only the responsibility of the Ministry of Health, the Ministry of Agriculture, or the Ministry of the Interior. This is a matter of governance; what should we do? What kinds of goods should we import, what kinds of goods should we produce, and what kinds of food products should we subsidize?"*
>
> *(P.14)*

The separation of public health and curative affairs has also challenged the referral system.

> *"Now, the referral system is also being discussed; where is it implemented? What types of referrals have been made? What was the effectiveness of this referral? Unfortunately, public health and curative have their own referral systems which are not connected." (P.4). "reforming the structure immediately will not assist, but if they can optimize the current structure, it may be more effective to meet the goals."*
>
> *(P.2)*

The NCDs management center and the public health deputy organizational structure must be strengthened. *"There are many limitations on the number of centers and offices for deputies. I believe that becoming a center rather than a disease office was preferable due to its greater authority. Proposals have been developed for a structure but have not been approved. This is notably true for tobacco, physical activity, and chronic respiratory disorders, which lack official, independent organizational structures" (P.10). "To battle cancer, we planned the administration to the National Cancer Management Center or the National Secretariat, but it was never put into practice" (P.23).*

Recommendation

- Examining the advantages and disadvantages of developing a health deputy as an alternative to two public health and curative deputies as a long-term intervention

- As a short-term measure, designing strategies for more effective engagement and cooperation between public health and curative deputies through the formation of joint working groups.

- Examining, approving, and developing the organizational structure of unstructured units (tobacco, physical activity, chronic respiratory).

- Implementation of the approved cancer office structure

- Examining the appropriate number of organizational positions for each unit

- Strengthen the organization of the SCHFS to enhance intersectoral cooperation.

### 3. Human resources management

The training of human resources at universities of medical sciences was not sufficiently community-oriented before the HTP; therefore, reforming the educational system is one of the phases considered for the HTP. This strategy considered ten country areas, and the education system was infused with essential disciplines and community-oriented education. *"In the field of preventive intervention, university-level training is very basic and not community-oriented. University-level training in the area of human resources does not meet the needs of society. In recent years, revisions have been made, and the outcomes of these modifications must be awaited" (P.8).* Future planning is somewhat more transparent and more manageable because different levels of service providers for diseases are defined. *"At the level of service provided, it has been seen what characteristics the human resources providing services at each level should have and what services they should provide (chronic respiratory diseases) (P.10).*

Communication skills are one of the issues that still exist between patients and providers; *"In particular, elderly patients have low literacy levels, and our service providers speak to them in, at best, simple scientific language, which frequently fails to communicate with this group of patients effectively. Effective communication skills training is essential for providers" (P.8).*

The ineffective referral and communication between curative and public health deputies have a negative impact on human production efficiency. *"...for example, if a diabetic patient with major symptoms goes directly to a specialist after his issue is cured, there is no follow-up; he should be sent to the PHC level, and the barriers that prevented him from being recognized by PHC must be addressed" (P.8).*

A shortage in human resources was also noted in related departments, particularly in the physical activity, cancer, and chronic respiratory units. "The next step is to add workforce after strengthening the structure" p10. "cancer affairs have many duties, at the national level, which cannot be managed by a five-person office" (P.23)

Following the implementation of the HTP, new services, such as nutrition and psychological counseling, are being provided in PHC centers facing difficulties. *"The number of population per nutritionist should be reduced, but this ratio is such that this expert cannot respond to the needs of the covered population with quality"* and *"we still do not have nutrition experts in most of our urban and rural centers"* and *"or In some centers, nutritionists still do not have computers"* p16 and *"the number of these forces is small compared to the workload" (P.23).*

Payment to providers is one of the most crucial policies contributing to maintaining and enhancing human resource performance. *"The current method of paying experts based on the number of services rendered is inaccurate, as the payments are less than the actual number of services provided and Prompted the departure of fed experts" (P.12).*

Units help to define the necessary training for medical and healthcare staff in PHC centers, but they are not particularly good at determining the educational needs of faculty and students at universities. It is essential to boost communication with the MoHME deputy of education to address this. *"There is almost no consultation with our office regarding the educational issues of university students and professors" (P.23).*

Recommendation

- Continuous feedback from units working on NCDs to the education system's planning system.

- Implementing patient-provider communication training

- Improving the referral system to maximize the utilization of human resources

## 4. Financial management

Managers and policymakers are receptive to reforms, innovations, and new sources of revenue and methods for allocating financial resources to improve financial management since they have extensive experience with the effects of reliance on financial resources on the instability of government revenues. As a result of the uncertainty of the health system's financing mechanisms, national plans have been designed with some confusion, and the quality of planning has declined.

> *"The financing status of the health system should adhere to a principle so that planners are aware of their responsibilities, whether public insurance or out of pocket; this situation has exposed our planners to a great deal of uncertainty"*

> *(P.4)*

The allocation of resources is currently facing significant challenges due to the adverse economic situation caused by sanctions and the decline in government public revenue. Also, several creative community-based projects were terminated. *"Before the tightening of sanctions, a limited budget had been set aside for the pilot study of chronic respiratory disorders, but in the previous two years, no funding has been received. (P.10).*

Chronic diseases come with everyday expenses that the patient must cover to survive the disease's serious effects. These factors should be considered during planning as well. *"The issue is that some people are ignoring their diabetes because managing it daily costs so much money"* (P.9).

Another issue that calls for capacity building is the Ministry of Health's duty to guide allocating financial resources outside the Ministry of Health. *"The Ministry of Health should supervise the allocation of resources for optimal nutrition, even though other ministries (other than the Ministry of Health) can do so. . . .. .the Ministry of Health should play a leading cross-sectoral role in the macro-politics of the country"* (P.14).

The absence of social and private insurance in preventive areas is one of the other issues. *"Many financial concerns will disappear for both individuals and insurance companies if insurance organizations support NCDs' preventive service packages"* (P.19).

Recommendation

- Clarifying for planners the future pathway of health system financing

- The departments of the Ministry of Health offer guidance and training for reporting programme outcomes and cost-effectiveness metrics.

- Investigating the construction of a comparable method agreed upon by the departments of the Ministry of Health to allocate resources among various initiatives.

- The Ministry of Health should pursue financing national health-related programs in areas such as agriculture, air pollution, etc.

- Insurance companies and the private sector covering preventative services

## 5. Programs management

The health system's transformation plan has impacted the service provider's organizational structure. Programs are gathered from various national units, including the maternal and newborn, environmental, communicable, and NCDs departments in the PHC network management center, and delivered as age-appropriate packages services that may not include all of a program's material. *"The HTP entirely changed the structure of the previous provider, and diabetes care is now delivered in the form of age groups. This poses challenges because the proposed program differs from the one currently in operation. It cannot be restricted to specific program components" (P.12).* Based on this, the units' programs must be delivered to the PHC network management center with the highest priority given to the age groups and service packages.

Implementing the integrated health system) SIB(, using electronic health records, the electronification of prescriptions for medications, and other initiatives have helped the country's health information system progress significantly in recent years. Nonetheless, there are no connected information systems between the various levels of service delivery.; *"Many issues will be resolved if the electronic health system is developed and adopted by everyone, including the private sector. This information system's fragmentation wastes the system's energy" (P.9).*

In the section on organizational structure, we highlighted the separation and poor communication of the curative and public health structures, which has also been reflected in the section on program management. *"Treatment-focused and prevention-focused initiatives should be connected" (P.1).*

Another area that calls for capacity building is the evaluation of programs. Comparing programs is another way to show how different effective programs are. *"Showing the results of programs isn't just about services that are provided; it is also about how many lives saved, how much costs saved, how disease burden reduced, and other things we don't have an obvious procedure to assess programs. "…. "For instance, a university that enacted a restriction on hookah supply throughout the entire area covered did not measure the impact on reducing tobacco usage, the impact on public support, or how satisfied people are with this policy." (P.5).*

Due to the country's hybrid health system, which includes components of several health systems worldwide, planning for Iran's health system has been challenging. "*Our healthcare system is extraordinarily complex and unclear. Politicians and planners face a severe challenge in this regard; they must develop a strategy that is compatible with all of the existing systems, as we utilize a combination of them" (P.8).*

The development of community-oriented programs should be included in the agenda to have the daily affairs of organizations and families; "Sometimes there are simple things that the universities of medical sciences themselves do not follow, for example, last year there was oil and sugar in the food basket donated to the employees of the university" (P.1)

Recommendation:

- Integrating preventative and treatment-oriented programs, as well as a strengthened referral and feedback mechanism between them

- Employing novel approaches and prioritizing interventions and programs in response to contractionary and expansionary economic situations.

- Developing the fundamental concepts of program evaluation so that the programs of various deputies and departments can be compared.

- Determining the future direction of the country's health system for planners (Buriji, Bismarck, out-of-pocket payment, national health insurance, etc.)

## 6. Relations management

*"The Secretariat of the Supreme Council is acting in the capabilities of inter-sectoral collaboration, but all departments of the Ministry of Health have the duty to move toward inter-sectoral cooperation" (P.17),* according to consensus among all departments that the Supreme Council of Health and Food Security is the best mechanism for fostering cross-sectoral measures.

*"When the national document of the NCD program was adopted, a large number of ministries also signed this document and committed to it, and this means attracting their participation, while we are in constant contact with other ministries through the Supreme Council of Health and Food Security"*

*(P.10).*

Executive body secretaries for health; It is a new idea that the Supreme Council implemented for Food and Health Safety. Based on this, a Ministry of Health representative has been placed in other ministries and strives to encourage inter-sectoral collaboration. There are currently 23 health secretariats, some of which are located in agriculture, sports, and education ministries.

The socio-economic commissions of the parliament provide an additional opportunity for the Ministry of Health to integrate health into all policies. *"The Ministry of Health should actively participate in parliamentary committees. When the annual budget is being approved, it should advocate for the expenditure of these funds in a manner that promotes a healthy diet. The Ministry of Health either does not perform certain functions or performs them inadequately"* (P.13).

The structure of the Ministry of Health also influences its intradepartmental relations or communications. *"Participating stakeholders in developing policies are, in my opinion, the most effective method for informing these policies; this will result in the key stakeholders of the adopted policy being automatically notified of the approval." (P.1)*. . . *"We do not have an optimal partnership with the treatment department, nor do we share the same information system" (P.12).*

Recommendation

- Expanding the duties and roles of the Ministry of Health through the Supreme Council of Health and relevant parliamentary commissions

- Improving intradepartmental communication and cooperation through the establishment of working groups, coordination committees

- The commitment and collaboration of the Ministry of Health to enhance health in all policies

## Discussion

Our study's findings served as a tool for identifying organizational capacity-building gaps in the areas we investigated. In the qualitative section, we outlined the root causes of these points and proposed interventions. Annex 3 of the Global Action Plan for the Prevention and Control of Noncommunicable Diseases has six objectives whose implementation will enable achieving NCDs objectives by 2030 [9]. One of the recommendations in Appendix 3, is capacity assessment and capacity building [18]; and countries have paid less attention to it [19].

To enhance healthcare conditions in developed and developing countries, capacity building is essential. This process focuses on understanding the barriers that prevent organizations

from realizing their goals and promoting those elements that aid them in achieving measurable and sustainable results [34, 35]. The evaluation of capacity building for NCDs can be viewed from various perspectives, including views of civil society organizations [36]. Different levels of capacity building are used for both organizations and individuals [37]. The capacity-building points may differ at the level of providers and local managers. This capacity assessment study was conducted at the national level and demonstrated the majority of the needs of national NCD managers.

Since capacity-building results and outputs are achieved over the long term (11), managers do not consider it as much as they should. Many studies on capacity assessment and capacity building have focused on the presence or absence of factors such as financial resources and medicines [20, 31, 38, 39] and less on the NCNCD's ability and authority to determine and guide factors. Some other capacity-building studies choose a specific dimension, such as the capacity assessment on the need for research and education for NCDs in Turkey [40].

The region's varied economic, cultural, and social development has influenced health policies; as a result, there are differences in the incidence of risk factors among provinces [41, 42]. Because of this different context, diseases, and risk factors, it is essential to modify national plans to local circumstances. For this reason, local health authorities need to have access to multidisciplinary decision-making teams that, under NCNCD leadership, adjust national agendas.

Clarifying the future direction of the health system's governance model in Iran for managers and policymakers and determining the ambiguous status of programs such as family physicians and basic insurance should be outlined for planners to decrease policymaking uncertainty. Similar research has emphasized the importance of the health system's governance model in developing action plans [43].

The results showed the consequences of the separation of public health and curative affairs in MoHME in human resources, resource allocation, and program management, Coherent governance. Experts' opinions on increasing integration vary; some believe that structural changes should be made, while others prefer alternative approaches. Structural reforms are costly to implement and will likely encounter opposition from stakeholders. Studies have noted the necessity to reform the Ministry of Health's organizational structure. For instance, organizational problems and scattered parallel structures in MoHME were explicitly mentioned by Behzad Damari et al. in 2020 [44], and Korosh Etemad in 2016 [45].

The Ministry of Health should have a goal to fight for when addressing health determinants, not just critical or advisory approaches. According to research on intersectoral cooperation on NCDs conducted in African nations, the main barriers to multisectoral action include the lack of awareness among various sectors of their potential contribution, a lack of political will, the difficulty of coordination, and a lack of resources [46]. Many of these can be carried out via developing capacity.

Many articles have talked about how important it is for countries to follow the WHO's Best Buys and other interventions, especially in countries with limited finances and resources. As this study and others have shown [47], Iran is in a good situation than other countries to set the agenda for these interventions [47–52]. The authority and capability of the key bodies should be reinforced to be implemented and indicators should also be used to measure the level of progress nationally and in each province.

Effective program implementation would be severely hampered by weak accountability and transparency in allocating resources, and health systems increasingly use a systematic approach to allocate resources based on evidence [53, 54]. The allocation of resources, which can increase the effectiveness of the Iranian health system, is one area that needs capacity building.

Inadequate intersectoral cooperation in many countries is one of the greatest challenges to addressing NCDs [55–57]. Fighting against NCDs as the leading cause of death in Iran and the world requires a high-level political commitment and a multi-sectoral approach in all countries; as a result, Iran established a multi-sectoral committee known as the Iran NCDs Committee (INCDC) and developed a national plan. The Supreme Council of Health and Food Safety has approved this multisectoral plan [58].

In this study, more than 27 policy interventions have been offered to build organizational capacity; some have umbrella status, meaning that changes to them might affect the entire system, including changing MoHME's organizational structure.

## Conclusion

The designed tool identified the need for organizational capacity in the field of NCDs, which can assist managers become more aware of the organizational challenges. The causes of organizational weakness and associated policy interventions are also listed in the qualitative section; we believe that these will serve as useful recommendations for managers and planners based on the available data.

## Supporting information

**S1 Appendix. Description of the expert teams and validity and reliability of the tool.**
(DOCX)

**S2 Appendix. Consolidated criteria for reporting qualitative studies (COREQ): 32-item checklist.**
(DOCX)

## Acknowledgments

The authors would like to thank the MoHME's authorities and staff. In particular, we are grateful to the NCNCD-MoHME, for their invaluable contribution to data collection and interpretation of findings.

## Author Contributions

**Conceptualization:** Ahad Bakhtiari, Amirhossein Takian.

**Data curation:** Ahad Bakhtiari, Masoud Behzadifar.

**Formal analysis:** Ahad Bakhtiari, Amirhossein Takian.

**Funding acquisition:** Ahad Bakhtiari.

**Investigation:** Ahad Bakhtiari.

**Methodology:** Ahad Bakhtiari, Amirhossein Takian.

**Project administration:** Ahad Bakhtiari, Amirhossein Takian.

**Resources:** Ahad Bakhtiari.

**Software:** Ahad Bakhtiari.

**Supervision:** Ahad Bakhtiari, Amirhossein Takian.

**Validation:** Ahad Bakhtiari.

**Visualization:** Ahad Bakhtiari.

**Writing – original draft:** Ahad Bakhtiari.

**Writing – review & editing:** Ahad Bakhtiari, Amirhossein Takian, Afshin Ostovar, Masoud Behzadifar, Efat Mohamadi, Maryam Ramezani.

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
