## [Decision Letter · Decision Letter 0]

24 Feb 2023

PONE-D-22-28106Developing an organizational capacity assessment tool and capacity-building package for the National Center for Prevention and Control of Noncommunicable Diseases in IranPLOS ONE

Dear Dr. Amirhossein Takian,

Thank you for submitting your manuscript to PLOS ONE. After careful consideration, we feel that it has merit but does not fully meet PLOS ONE’s publication criteria as it currently stands. Therefore, we invite you to submit a revised version of the manuscript that addresses the points raised during the review process.

We look forward to receiving your revised manuscript.

Kind regards,

Samane Shirahmadi, PhD

Academic Editor

PLOS ONE

Journal Requirements:

“AT and AO are members of the INCDC at the MoHME- Iran. We (AT and AO) have no conflict of interest to disclose. AB, , EM, MB, and MR declare that they have no competing interests.”

We note that one or more of the authors are employed by a commercial company: INCDC at the MoHME- Iran

4**. **Your ethics statement should only appear in the Methods section of your manuscript. If your ethics statement is written in any section besides the Methods, please move it to the Methods section and delete it from any other section. Please ensure that your ethics statement is included in your manuscript, as the ethics statement entered into the online submission form will not be published alongside your manuscript.

Reviewers' comments:

Reviewer's Responses to Questions

**Comments to the Author**

1. Is the manuscript technically sound, and do the data support the conclusions?

Reviewer #1: No

Reviewer #2: Yes

Reviewer #3: Partly

2. Has the statistical analysis been performed appropriately and rigorously? 

Reviewer #1: No

Reviewer #2: Yes

Reviewer #3: No

3. Have the authors made all data underlying the findings in their manuscript fully available?

Reviewer #1: No

Reviewer #2: Yes

Reviewer #3: No

4. Is the manuscript presented in an intelligible fashion and written in standard English?

Reviewer #1: No

Reviewer #2: Yes

Reviewer #3: Yes

5. Review Comments to the Author

Reviewer #1: Capacity assessment tools are increasingly, and effectively, used by health systems around the globe to assess and improve their capacities to respond to public health emergencies and other problems. Given the prominence of non-communicable diseases, a similar approach to NCD capacities, as described in this paper, is a promising idea. If it were validated, the authors’ tool could be useful in many countries beyond Iran, where it was developed and tested.

However, despite the title (“Developing an organizational capacity assessment tool …”), the parts of paper addressing the performance of the proposed tool are poorly described and I believe under-developed. The methods for assessing the tool’s validity and reliability are given in one-paragraph (Section 2.1) and the related results in another paragraph (the first paragraph in the Results section). Appendix A has only two paragraphs (those immediately preceding Table 2) on methods and results.

Overall, I find these paragraphs difficult to follow, both because of substantive reasons and also poor writing. In particular:

• Section 2.1 describes two expert teams, noting that “For the first expert team (18 experts), we selected 12 experts …” The second team has 6 experts, so perhaps that’s the difference, but it’s very hard to follow.

• More substantively, these teams are described in different ways in the main paper and the appendix, but I believe that some work within the relevant ministries and others not. If this is true, the first would be evaluating their own work, so there is a potential for bias. It would be interesting to see if both categories of experts had similar or different ratings, but it is not clear to me that this is done.

• The paper describes formal interviews with the six experts from the second team, but it is not clear whether or how they were used to assess the validity or reliability of the tool. Presumably the discussions on pp. 9-16 is based on these interviews, but these seem to be substantive, and unrelated to assessing the tool’s performance.

• The paragraph in the Results section refers to Appendix A, Table 1-6. The appendix has only 2 numbered tables, so I can only guess that “Table 1-6” refers to the six un-numbered domain-specific tables.

• Referring to “Table 2-Appendix A”, the Results section reports on “kappa values for all of the arias (areas?) we tested show, but one see only in the appendix that that the results are based on only three substantive areas (physical activity, cardiovascular disease, and tobacco).

Because of the lack of clarity about methods and results, it is difficult for me to know that the tool is valid and performs well even in the setting in which it was tested.

Rather than validating the tool they developed, the authors spend the bulk of the paper in substantive discussion of the six domains (pp. 9-16), providing specific recommendations for organization change. It is not clear to me that these discussions are based on the questionnaire results; rather they seem to be based on the qualitative interviews. The Discussion and Conclusions (pp. 17-18) also primarily address substantive issues. As a methodologist who is not familiar with Iran, I have no way to assess the validity of this discussion. But even if they were valid, they would be of little interest to researchers or practitioners outside of Iran.

Reviewer #2: Thanks for choosing me as a reviewer.

The article is very valuable and constructively written, however, the following suggestions are made for improvement.

The study was written using a structure that made sense.

Introduction: Given the high prevalence of NCDs noted in the introduction, could you perhaps elaborate on how these conditions interact with the COVID-19 pandemic? Make an effort to use numbers to demonstrate the subject's importance.

More information can be found in the WHO research at https://www.who.int/publications-detail-redirect/9789240010291.

Even when the questions on the COREQ checklist have been addressed, it is still advised to complete and submit this checklist. The table title ("Reported on Page #") invites the authors to provide the page numbers that correspond to each specific item on the checklist.

Best Regards

Reviewer #3: Thank you for giving me the opportunity the reading this valuable research. In this research, a tool has been designed to assess the capacity of the National Center for Non-Communicable Disease for Non-Communicable Disease and in another part of the study; the NCNCD organizational capacity has been assessed. The design of the study is mixed method and it had qualitative and quantitative parts.

Here are some comments:

Abstract:

- Please do not use the abbreviations in the abstract.

- Please describe what you had done in quantitative and qualitative sections of your study in clear form in design section, separately.

- What is the type of your mixed method design (exploratory? explanatory?)?

- How do you use and combine the findings of the two parts of the study? Describe it in the method and result.

- The conclusion did not cover all the findings of your study.

Introduction

- The knowledge gap is not well discussed in the introduction section.

- you should report the previous related tool in your research topic and why you want design a new tool.

- The goal is not clear well.

Method

- What is the type of your mixed method design (exploratory? explanatory?)? why you select mixed method for your research? Please describe it. What is the qualitative phase and what is the quantitative phase and how this two phases are related and integrated?

- It is not clear what are the goal of literature review (step 1 in method) and what is the relation of this step to other steps of the study. Is your goal is item generation for the tool?

- 2. Designing Tool (Organizational Capacity Assessment Tool, OCAT):

is there related tool previously?

what was your search strategy? how did you check the quality of studies? what were your keywords? what document you have searched for? what language did your document have? what was your time limits? what studies had been searched? what were your inclusion and exclusion criteria for the studies? you should draw the PRISMA checklist for your research.

why did you checked only CVI? why didn’t check other validity properties? why did you select 18 experts? how do you calculate CVI? why did you select only 6 experts for relability assessment? how did you check the inter-rater agreement and which coefficient did you calculate?

which type of the tool did you design? (checklist, questionnaire, scale,…)

- The descriptions which you have provided in the 3.1 and 3.3 section, are related to tool designing step instead of these two steps, you could provide one step for data gathering via the tool which have been designed in the previous section.

- Interviews with NCNCD and expert team two:

How did you insure data saturation? What is the method of thematic analysis? Did the interview were recorded? How did you checked the rigor of the qualitative phase of your study?

- Data analysis: please explain more details about thematic content analysis with reference.

Result:

- You reported the CVR in tables, but you didn’t describe the assessment method of it in the method section.

- You should report how much item did you generate at the first, and how much items were deleted in the process of content validity assessment.

- You shoud describe how and which Kappa did you assessed in the method section and then report the result only in the result section.

- you should report how many codes, subcategories, main categories and themes did you gathered. how was the process of abstraction of the finding (it is better to report this process as a table or figure)

- You should show how the findings of the two part of your study are related to each other in the finding section, too.

- you should report the analysis of the findings which you gathered with the tool.

Discussion:

- In the first line and paragraph of the discussion section, conclude the goal of your study and the main findings.

- What was your limitations? What is your further researches recommendations? What was your findings implication for practice?

Conclusion

- The conclusion did not cover all the findings of your study.

Appendix:

- the table did not have numbers.

6. PLOS authors have the option to publish the peer review history of their article (what does this mean?). If published, this will include your full peer review and any attached files.

Reviewer #1: **Yes: **Michael A Stoto

Reviewer #2: **Yes: **Dr Samad Azari

Reviewer #3: No

---

## [Author Response · Author response to Decision Letter 0]

3 May 2023

April 20, 2023

Dear editors;

PLOS ONE

Re: PONE-D-22-28106

Developing an organizational capacity assessment tool and capacity-building package for the National Center for Prevention and Control of Noncommunicable Diseases in Iran

Dear Editor,

Thank you and the reviewers very much for your recent comments, which provided us

the opportunity to improve our manuscript. We are pleased to inform you that we have

addressed all the comments in in order that they raised, as you will find them below.

Comments to the Author

Reviewer #1: 

1. Capacity assessment tools are increasingly, and effectively, used by health systems around the globe to assess and improve their capacities to respond to public health emergencies and other problems. Given the prominence of non-communicable diseases, a similar approach to NCD capacities, as described in this paper, is a promising idea. If it were validated, the authors’ tool could be useful in many countries beyond Iran, where it was developed and tested.

Re: Thank you. Non-communicable diseases are the leading cause of premature death, as you mentioned, so researchers, managers, and policymakers must pay close attention to this issue. In WHO statements, this need has been referred to numerous times. The Validity section has been revised to accommodate this comment.

2. However, despite the title (“Developing an organizational capacity assessment tool …”), the parts of paper addressing the performance of the proposed tool are poorly described and I believe under-developed. The methods for assessing the tool’s validity and reliability are given in one-paragraph (Section 2.1) and the related results in another paragraph (the first paragraph in the Results section). Appendix A has only two paragraphs (those immediately preceding Table 2) on methods and results. Overall, I find these paragraphs difficult to follow, both because of substantive reasons and also poor writing. In particular:

• Section 2.1 describes two expert teams, noting that “For the first expert team (18 experts), we selected 12 experts …” The second team has 6 experts, so perhaps that’s the difference, but it’s very hard to follow.

• More substantively, these teams are described in different ways in the main paper and the appendix, but I believe that some work within the relevant ministries and others not. If this is true, the first would be evaluating their own work, so there is a potential for bias. It would be interesting to see if both categories of experts had similar or different ratings, but it is not clear to me that this is done.

• The paper describes formal interviews with the six experts from the second team, but it is not clear whether or how they were used to assess the validity or reliability of the tool. Presumably the discussions on pp. 9-16 is based on these interviews, but these seem to be substantive, and unrelated to assessing the tool’s performance.

Re: Thank you for your comments.

One of the crucial steps in the development of a capacity assessment tool is, as you mentioned, ensuring its validity and reliability. Perhaps because we combined three phases (mentioned below); tool's reliability and validity is not stated clearly: 

• Design and validity of the tool

• Tool scoring by stakeholders and the results

• Discussing challenges, opportunities, and solutions using the qualitative approches

Each of these might stand alone as a separate article.

This study is the third part of a larger study, the first two parts of which have been published and can be found at the link below.

1. Assessment and prioritization of the WHO “best buys” and other recommended interventions for the prevention and control of non-communicable diseases in Iran, (Link)

2. Intersectoral collaboration in the management of non-communicable disease’s risk factors in Iran: stakeholders and social network analysis (Link)

We used multiple expert teams during various phases of study

• During tool design (expert team number one) (Table 1 appendix A) 

• During tool validity and reliability (expert team number two) (Table 2 appendix A)

• While identifying challenges and opportunities (qualitative phase-expert team number three) (Table 3 appendix A).

Cognitive interviews, CVI, and CVR were used to assess the validity. We added the profiles of experts who participated in the validity and reliability of the tool. The validity and reliability section was revised in the main text and appendix A. Please see to the appendix-A and main text's validity sections. We selected 12 experts for validity and 3+6 experts for reliability.The kappa values show moderate (0.4–0.6) to strong (0.6 and higher) interrater agreement. In the designed tool, experts assign a score to each of the sub-dimensions or questions. Tables 6–11 of the appendix A show the degree of validity for each subdomain. Appendix A-Table 5 shows the reliability values. Many capacity assessment tools use the Kappa test for reliability measurement. Please see the link below for more information.

Link : https://www.ncbi.nlm.nih.gov/pmc/articles/PMC3900052/

Self-assessments has been widely used in the past for measuring organizational capacity. The evaluated organization's human resources (experts and managers) examine the capacity of the organization to fulfill its duties; and the capacity of the organization is measured rather than individual performance. Examples can be found in the links below.

1. https://usaidlearninglab.org/resources/organizational-capacity-assessment (Description section)

2. https://www.ngoconnect.net/sites/default/files/resources/Organizational%20Capacity%20Self-Assessment%20Tool%20-%20Training%20Guidelines.pdf

3. https://www.jsi.com/resource/organizational-capacity-assessment-oca-tool-participants-copy/

As you can see, many tools have a common main domain, and in the initial cycle of capacity assessment, subdomains focus on the key areas. Subdomain questions may alter in the upcoming cycle of capacity assessment because it is a continuous process.

The boundaries between quantitative and qualitative phases became more clear. This is an explanatory sequential design that has a quantitative phase and a qualitative phase.

3. The paragraph in the Results section refers to Appendix A, Table 1-6. The appendix has only 2 numbered tables, so I can only guess that “Table 1-6” refers to the six un-numbered domain-specific tables.

Re: Thank you. Every table's title in Appendix A has been revised. Please see Appendix-Tables 6, 7, 8, 9, 10, and 11. Additionally, based on the 12 experts, this Tabels reported the validity (CVI and CVR) of each subdomain. 

4. Referring to “Table 2-Appendix A”, the Results section reports on “kappa values for all of the arias (areas?) we tested show, but one see only in the appendix that that the results are based on only three substantive areas (physical activity, cardiovascular disease, and tobacco).

Re:Thank you. Revised. Since the framework, questions, and scoring processes in domains and subdomains are the same in all 7 areas, the study team believed that repetition in one area (two experts) is a more important criterion for measuring reliability. Therefore, we run the test in these three areas. Twice in three area (two experts separately for each). 

5. Because of the lack of clarity about methods and results, it is difficult for me to know that the tool is valid and performs well even in the setting in which it was tested.

Re: We revised the method and findings sections and hope that they will be convincing to the reviewers. Please see the Method and Results sections and Appendix A. We are pleased tominform you that the Ministry of Health & Medical of Iran has put some of our recommended interventions (qualitative phase) on agenda, including strengthening the organizational structure.

6. Rather than validating the tool they developed, the authors spend the bulk of the paper in substantive discussion of the six domains (pp. 9-16), providing specific recommendations for organization change. It is not clear to me that these discussions are based on the questionnaire results; rather they seem to be based on the qualitative interviews. The Discussion and Conclusions (pp. 17-18) also primarily address substantive issues. As a methodologist who is not familiar with Iran, I have no way to assess the validity of this discussion. But even if they were valid, they would be of little interest to researchers or practitioners outside of Iran.

Re:We hope the revisions we made have been able to address your concerns. Pages 9–16 contain information from the study's qualitative phase. The qualitative phase was carried out in order to closely examine the organizational weaknesses (as determined by the tool). To make the distinction clear to readers, we changed the headings and defined the boundaries between the quantitative and qualitative phases.

The recommended interventions in the findings and discussion section seem more appropriate for developing countries, where the structures are not yet fully developed and there are severe financial and human resource shortages.

Reviewer #2: 

1. Introduction: Given the high prevalence of NCDs noted in the introduction, could you perhaps elaborate on how these conditions interact with the COVID-19 pandemic? Make an effort to use numbers to demonstrate the subject's importance.

More information can be found in the WHO research at https://www.who.int/publications-detail-redirect/9789240010291.

Re: Thank you for your thoughtful comment. Unfortunately, the COVID-19 crisis caused significant disruption in the delivery of NCD prevention and treatment services. Furthermore, many of those who died as a result of COVID-19 had NCDs or related risk factors. This was mentioned briefly in the introduction. Please see paragraph 3 on page 3.

2. Even when the questions on the COREQ checklist have been addressed, it is still advised to complete and submit this checklist. The table title ("Reported on Page #") invites the authors to provide the page numbers that correspond to each specific item on the checklist.

Re: Thank you. We fulfilled the check list for the study's qualitative parts as a Appendix B. Please see Appendix B

Reviewer #3: 

Abstract:

1.Please do not use the abbreviations in the abstract.

Re: Thank you; we revised the abstract. Please see abstract.

2. Please describe what you had done in quantitative and qualitative sections of your study in clear form in design section, separately.

- What is the type of your mixed method design (exploratory? explanatory?)?

Re: Revised in abstract and method section please see page 2-abstract-method and page 4 method section; first paragraph.

3. How do you use and combine the findings of the two parts of the study? Describe it in the method and result.

Re: Please see abstract-method, the first paragraph of the method, the first paragraph of 4. Qualitative phase, and the first paragraph of each DOMAIN in the result section.

4. The conclusion did not cover all the findings of your study.

Re: Revised, please see abstract- conclusion

Introduction

5. The knowledge gap is not well discussed in the introduction section.

- you should report the previous related tool in your research topic and why you want design a new tool.

- The goal is not clear well.

Re: Done. Please see the last paragraph of the introduction.

Method

6. What is the type of your mixed method design (exploratory? explanatory?)? why you select mixed method for your research? Please describe it. What is the qualitative phase and what is the quantitative phase and how this two phases are related and integrated?

Re: Revised in abstract and method section please see page 2-abstract-method and page 4 method section; first paragraph.

7. It is not clear what are the goal of literature review (step 1 in method) and what is the relation of this step to other steps of the study. Is your goal is item generation for the tool?

Re: One of the first steps in capacity assessment and capacity building, which is recommended in many tools, is the initial familiarization with the dimensions of capacity assessment, a general knowledge of the related subject (here NCDs) and a basic knowledge of the organization under investigation.

As a step in that direction, this has been done.

8. Designing Tool (Organizational Capacity Assessment Tool, OCAT):

is there related tool previously?

Re: There is no comprehensive tool available in the field of NCDs for measuring the organizational capacity of the institution in charge of this topic.

 There is a survey questionnaire in the topic of national capacity with different dimensions from the one created for this study. And it does not emphasize organizational capability

The WHO questionnaire inquires about the existence of a response strategy for a disease, while our tool looks for the team responsible for developing the strategy and the degree of adherence to it.

While our tool asked questions about the approved organizational structure, parallel institutions, and the institution's position in the overall structure, the WHO questionnaire only inquired about the existence of an institution for NCDs within the ministry. The WHO survey questionnaire was reviewed at as one of the key documents, and its most crucial points were covered in more depth in the tool than the organizational capacity aspect.

Please see the following link.

Link click here

9. what was your search strategy? how did you check the quality of studies? what were your keywords? what document you have searched for? what language did your document have? what was your time limits? what studies had been searched? what were your inclusion and exclusion criteria for the studies? you should draw the PRISMA checklist for your research.

Re: Revised. Please see Appendix A-Box B

10. Why did you checked only CVI? why didn’t check other validity properties? why did you select 18 experts? how do you calculate CVI? why did you select only 6 experts for relability assessment? how did you check the inter-rater agreement and which coefficient did you calculate?

which type of the tool did you design? (checklist, questionnaire, scale,…)

Re: Cognitive interviews, CVI, and CVR were used to assess the validity. We added the profiles of experts who participated in the validity and reliability of the tool. The validity and reliability section was revised in the main text and appendix A. Please see to the appendix-A and main text's validity sections. We selected 12 experts for validity and 3+6 experts for reliability.The kappa values show moderate (0.4–0.6) to strong (0.6 and higher) interrater agreement. In the designed tool, experts assign a score to each of the sub-dimensions or questions.

11. The descriptions which you have provided in the 3.1 and 3.3 section, are related to tool designing step instead of these two steps, you could provide one step for data gathering via the tool which have been designed in the previous section.

Re: Revised. Please see section 3.1 . first paragraph

12. Interviews with NCNCD and expert team two:

How did you insure data saturation? What is the method of thematic analysis? Did the interview were recorded? How did you checked the rigor of the qualitative phase of your study? 

- Data analysis: please explain more details about thematic content analysis with reference.

Re:. Revised please see section 4. Qualitative phase. After including the expert teams from earlier phases, the third expert team is now relevant for this phase.

Result:

13. You reported the CVR in tables, but you didn’t describe the assessment method of it in the method section.

Re: revised please see validity section of method.

14. You should report how much item did you generate at the first, and how much items were deleted in the process of content validity assessment.

Re: Thank you. Revised please see appendix A section: Validity . ‘’Three of the 18 sub-dimensions or tool questions were modified and their validity was assessed again; including subdomains 3.3; 5.1 and 5.3)’’

15. You shoud describe how and which Kappa did you assessed in the method section and then report the result only in the result section.

Re: Please see Appendix A section of reliability.

16. you should report how many codes, subcategories, main categories and themes did you gathered. how was the process of abstraction of the finding (it is better to report this process as a table or figure)

Re: Revised please see page 8. “Six main themes (domains) and 18 sub-themes (subdomains) were utilized for arranging 438 codes’’ 

17. You should show how the findings of the two part of your study are related to each other in the finding section, too.

Re: Thank you. Revised. Please see the first paragraph on page 10

18. you should report the analysis of the findings which you gathered with the tool.

Re: Thank you. Revised. Please see the last paragraph on page 8

Discussion:

19. In the first line and paragraph of the discussion section, conclude the goal of your study and the main findings.

Re: Thank you. Revised. Please see first line and paragraph of the discussion section.

20. What was your limitations? What is your further researches recommendations? What was your findings implication for practice?

Re: As previously stated, capacity assessment and capacity building are dynamic processes that must be repeated after the initial intervention. For example, with the changes that are likely to occur probably in the structure of the Ministry of Health, the capacity assessment must be repeated.

Conclusion

21. The conclusion did not cover all the findings of your study.

Re: Revised. Please see Conclusion.

Appendix:

22. the table did not have numbers.

Re: Thank you. Revised. Please see Appendix A.

Again, thank you very much for providing us with the opportunity to improve our work. We hope that you and your team will find the revisions up to your satisfaction and look forward to your decision in due course.

Yours sincerely,

Amirhossein Takian, MD MPH Ph.D. FHEA

Professor and Head, Department of Global Health and Public Policy

School of Public Health, Tehran University of Medical Sciences, Tehran, Iran

Corresponding author

---

## [Decision Letter · Decision Letter 1]

14 Jun 2023

Developing an organizational capacity assessment tool and capacity-building package for the National Center for Prevention and Control of Noncommunicable Diseases in Iran

PONE-D-22-28106R1

Dear Dr. Amirhossein Takian ,

We’re pleased to inform you that your manuscript has been judged scientifically suitable for publication and will be formally accepted for publication once it meets all outstanding technical requirements.

Kind regards,

Samane Shirahmadi, PhD

Academic Editor

PLOS ONE

Additional Editor Comments (optional):

Reviewers' comments:

Reviewer's Responses to Questions

**Comments to the Author**

1. If the authors have adequately addressed your comments raised in a previous round of review and you feel that this manuscript is now acceptable for publication, you may indicate that here to bypass the “Comments to the Author” section, enter your conflict of interest statement in the “Confidential to Editor” section, and submit your "Accept" recommendation.

Reviewer #2: All comments have been addressed

Reviewer #3: All comments have been addressed

2. Is the manuscript technically sound, and do the data support the conclusions?

Reviewer #2: Yes

Reviewer #3: Yes

3. Has the statistical analysis been performed appropriately and rigorously? 

Reviewer #2: Yes

Reviewer #3: Yes

4. Have the authors made all data underlying the findings in their manuscript fully available?

Reviewer #2: Yes

Reviewer #3: Yes

5. Is the manuscript presented in an intelligible fashion and written in standard English?

Reviewer #2: Yes

Reviewer #3: Yes

6. Review Comments to the Author

Reviewer #2: (No Response)

Reviewer #3: please edit the appendix A: some numbers are written in Farsi language.

All the comments have been addressed

7. PLOS authors have the option to publish the peer review history of their article (what does this mean?). If published, this will include your full peer review and any attached files.

Reviewer #2: **Yes: **Samad Azari

Reviewer #3: No

---

## [Editor Report · Acceptance letter]

20 Jun 2023

PONE-D-22-28106R1 

Developing an organizational capacity assessment tool and capacity-building package for the National Center for Prevention and Control of Noncommunicable Diseases in Iran 

Dear Dr. Takian:

I'm pleased to inform you that your manuscript has been deemed suitable for publication in PLOS ONE. Congratulations! Your manuscript is now with our production department. 

Kind regards, 

on behalf of

Dr. Samane Shirahmadi 

Academic Editor

PLOS ONE